Unveiling the clinical profiles of critically ill COVID-19 patients: insights from Ezhou during the early spread

Yang Guohui 1
Liu Zewen 2
Abraham Tabitha 3
Li Linwei 3
Zhou Tingyang 4
Zhang Qing zhangqing198411@126.com 5
Zuo Li zuo.4@outlook.com 6
1 Department of Intensive Medicine, The Affiliated Hospital of Guizhou Medical University , Guiyang , Guizhou , China
2 Department of Anesthesiology, Affiliated Ezhou Central Hospital of Wuhan University , Ezhou , China
3 The University of Texas Rio Grande Valley School of Medicine , Edinburg , TX , United States of America
4 The Interdisciplinary Biophysics Graduate Program, The Ohio State University , Columbus , OH , United States of America
5 Department of Intensive Medicine, Affiliated Ezhou Central Hospital of Wuhan University , Ezhou , China
6 Department of Physical Therapy, Medical Education, and Neuroscience, The University of Texas and UT Health Rio Grande Valley, School of Medicine, College of Health Professions , Edinburg , TX , United States of America
Teh Cindy Shuan Ju
Electronic publication date: 2025 Nov 20
Publication date: 2025
Volume: 13
Electronic Location ID: e20318
Received 2024 Aug 16; Accepted 2025 Oct 9
Copyright: ©2025 Yang et al.
Copyright year: 2025
Copyright holder: Yang et al.
License: This is an open access article distributed under the terms of the Creative Commons Attribution License, which permits unrestricted use, distribution, reproduction and adaptation in any medium and for any purpose provided that it is properly attributed. For attribution, the original author(s), title, publication source (PeerJ) and either DOI or URL of the article must be cited.
License URL: https://creativecommons.org/licenses/by/4.0/

Keywords: SARS-CoV-2, Clinical characteristics, High-resolution CT (HRCT), Mortality

Funding: Natural Science Foundation of Hubei Province No. 2018CFB677 This work was supported by Natural Science Foundation of Hubei Province (No. 2018CFB677). The funders had no role in study design, data collection and analysis, decision to publish, or preparation of the manuscript.

==============================
The coronavirus disease 2019 (COVID-19) started in December 2019 in Wuhan. This article evaluated clinical characteristics, and imaging manifestations in Ezhou, an early locked-down city, 80 kilometers east of Wuhan. We analyzed data from 98 confirmed severe COVID-19 patients in Ezhou Central Hospital between February 1st and March 22nd, 2020. We compared patients’ laboratory results, imaging manifestations, and treatments between survival and death groups. Of these 98 confirmed COVID-19 patients, 24 individuals (24.6%) had chronic diseases. The main symptoms of critically ill patients were fever (86.7%), cough (83.7%), and dyspnea (59.1%). Common complications were acute respiratory distress syndrome (ARDS; 49%), acute kidney injury (37.7%), and multiple organ dysfunction syndrome (MODS; 32.6%). Computed tomography (CT) scans displayed ground-glass opacity at subpleural regions that were associated with interlobular septal thickening. Within 28 days, 39 (39.8%) patients died. Compared to survivors, the death group had a higher median age (69.8 vs. 61.3, p < 0.05), and were more prone to ARDS (100% vs. 15.2%) and MODS (76% vs. 3.4%). Our report showed that in the early days of the COVID-19 outbreak, there was a high mortality rate in critically ill patients. Elderly patients (>65 years) normally have an increased risk of complications and ARDS. Most non-survivors were highly dependent upon mechanical ventilation. CT scans with imaging manifestations showed abnormal conditions in the lower multiple bilateral lung lobes, which provides a useful characterization of this fatal disease by recognizing COVID-19 pneumonia and assessing its evolution for the target for intervention of the patient recovery.

Introduction

The coronavirus disease 2019 (COVID-19) is a newly discovered global infectious disease, spreading rapidly from Wuhan and its adjacent city of Ezhou (80 kilometers from Wuhan) to other regions of China since February 2020. Compared to influenza, COVID-19 has a higher rate of infectiousness and mortality. Most severely ill patients require hospitalization (Li et al., 2020b).

COVID-19 is clinically classified as having four stages: mild, moderate, serious, and severe or critical forms. The virus invades lower respiratory tracts and can induce a massive release of inflammatory factors such as IL-6, and TNF-α, causing a “cytokine storm” with prolonged inflammation (Mirzaie et al., 2020). Severe acute respiratory syndrome coronavirus 2 (SARS-CoV-2) invades the human body through angiotensin-converting enzyme 2 (ACE2). Since ACE2 is well distributed in all major human organs, SARS-Cov-2 can damage the lungs, as well as the heart, kidneys, and other important organs. Various factors including inflammatory storm and multiple organ damage can lead to a high mortality in severely ill patients (Gupta et al., 2020; Inciardi et al., 2020; Robbins-Juarez et al., 2020; Shi et al., 2020). About 20% of the patients require hospitalization (Pormohammad et al., 2020; Wu & McGoogan, 2020). Although the mortality rate of COVID-19 is lower than that of Severe Acute Respiratory Syndrome (SARS) and Middle East Respiratory Syndrome (MERS) (Lu et al., 2020; Petrosillo et al., 2020), COVID-19 has exerted tremendous pressure on medical systems due to its extremely large number of infections (Havers et al., 2020; Korth et al., 2020).

Alongside the use of COVID-19 vaccines, it is crucial to effectively treat critically ill COVID-19 patients and thus reduce mortality. Previous studies well described the general epidemiological findings and clinical outcomes of COVID-19 patients (Liu et al., 2020; Tian et al., 2020; Wiersinga et al., 2020). The objective of our study is to further present clinical characteristics, analyze imaging manifestations to detect lung abnormalities and report the outcomes of severe cases during the early spread of COVID-19.

Furthermore, our research does not aim to report recent updates in COVID-19 or any mutated strains that dominate the global landscape today. Instead, our focus is on the early phase of the pandemic, precisely the period immediately following the Wuhan outbreak, when the virus was still in its original form. This is a critical period that may seem distant from the current format of COVID-19. However, our understanding of how the virus presented and affected critically ill patients at the early stage of the epidemic remains highly valuable in a nearby region of Ezhou with limited prior research; this insight is essential for historical perspective and understanding of how the disease clinically evolved. This research provides valuable information to medical professionals to recognize COVID-19 pneumonia and assess its evolution, which can help better characterize this fatal disease at its early stages.

Materials and methods

Research background

Our study was conducted at Ezhou Central Hospital, which was nationally designated as a center for treating severe and critically ill patients with COVID-19. A retrospective analysis was performed on data based on patients diagnosed with COVID-19 from February 1, 2020, to March 20, 2020. This period was selected, because it was in the early spread of COVID-19 before new treatments were implemented. Our study was approved by the ethics committee of the Ezhou Central Hospital (IRB#L2020-Y-012). All patients were diagnosed by the local health department for COVID-19 infection. We identified critically ill patients and analyzed their documents (admission records, electronic medical records, and corresponding patient care resources).

Diagnostic and exclusion criteria

All patients met the requirements of “Diagnosis and Treatment Protocol for Novel Coronavirus Pneumonia of the National Health and Welfare Commission (Trial Version 6)” (Xu, Chen & Tang, 2020). As Ezhou Central Hospital was the designated center for critically ill patients, a natural referral bias may exist, as patients with severe diseases were prioritized for admission. While this may limit generalizability to mild or moderate cases, our study mainly focused on critically ill patients to provide insights into this vulnerable subgroup during the early stages of the pandemic in a nearby heavily populated city such as Ezhou. Patients with COVID-19 who met one of the following conditions were defined as critically ill:

• Occurrence of shock

• Respiratory failure with the need for mechanical ventilation

• Combining the preceding with organ failures that require ICU monitoring treatments.

We excluded the following:

• Patients with autoimmune diseases, immunodeficiency diseases, or hypersensitivity diseases.

• Patients with malignant tumors who had received radio-chemotherapy.

• Patients taking hormones and immunomodulators within the past three months.

• Pregnant women

• Individuals with a negative reverse transcription polymerase chain reaction (RT-PCR) test for COVID-19 via throat swabs.

Grouping

Critically ill patients with COVID-19 were divided into two major groups: the survival and the death groups. For the survival group, both patients’ clinical symptoms and laboratory examination results suggested improvements compared to the death group. For the death group, both patients’ clinical symptoms and laboratory examination indicators worsened and died within 28 days. Thus, we focused on 98 severe cases. We divided the cases into two major groups: the survival and the death groups. The survival group had 59 patients (38 males and 21 females, aged 40–76 years, with an average age of (61.3 ± 13.4) years). The death group had 39 patients (26 males and 13 females, aged 45–89 years, with an average age of (66.8 ± 12.4) years). We examined the patient’s clinical symptoms and laboratory examination results. We found that survival group’s clinical symptoms and laboratory examination results revealed several signs that indicated a higher chance of longevity.

Data collection

We reviewed the clinical electronic medical records, admission data, laboratory examination values, and imaging results from all confirmed COVID-19-infected patients. The data were evaluated and collected using the acute respiratory distress syndrome guideline (Fan et al., 2017). Any missing or uncertain records were clarified through direct communications with medical service providers and their families. We collected the following data:

• General information: age, gender, exposure history

• Chronic medical history: chronic heart disease, chronic lung disease, cerebrovascular disease, chronic neurological disease, diabetes

• Symptoms: fever, cough, breathing difficulty, arthralgia, chest pain, headache, vomiting, diarrhea

• Vital signs: respiratory rate, pulse, blood pressure at the time of admission to the ICU

• Laboratory examination results upon the time of admission: hemoglobin, white blood cell count, lymphocyte count, platelets, albumin, etc.

Statistics

We presented descriptive data as the mean (or ± SD), or numerical values of variables expressed as percentages (%). Based on the parametric or nonparametric data of continuous variables and Fisher’s exact test for categorical variables, the difference between survival and death groups was evaluated using the double sampling t-test or Wilcoxon rank-sum test. We used the Kaplan–Meier chart for the generated data. The test was bidirectional, and the significance was set to α less than 0.05 (p < 0.05). All data were analyzed using Stata / IC15.1 software (Stata Corp, University City, Texas, USA).

Results

Demographic and baseline characteristics of patients with severe COVID-19

During the early spread of COVID-19, as of March 20, 2020, we had a total of 880 patients who had been diagnosed with COVID-19. A total of 782 (88.8%) of them were non-critical type, and 98 (11.1%) were critically ill patients as we listed in the sections above for the focus of the current study. The patient demographic characteristics are present in Table 1. Among the 98 severe cases, the average age was 64.6 ± 13.6 years old, of which 68 (69.3%) were over 60. 64 (65.3%) patients were male, and 34 (34.7%) patients were female. Twenty-four patients (24.6%) had chronic diseases, nine patients (9.2%) had respiratory diseases, eight patients (8.1%) had cardiovascular diseases, and five patients (5.2%) had cerebrovascular diseases (Table 1).

Table 1 Demographic characteristics and distributions of severe patients with COVID-19.

	Survival (n = 59)	Death (n = 39)	Total (n = 98)	
Age, yr	61.3 ± 13.4 (SD)	66.8 ± 12.4 (SD)*	64.6 ± 13.6 (SD)	
Age range distribution				
40–49	10 (17%)	2 (5%)	12 (12.4%)	
50–59	14 (24.7%)	4 (105)	18 (18.3%)	
60–69	20 (34%)	8 (20%)	28 (28.6%)	
70–79	13 (22%)	18 (46%)	31 (31.6%)	
≥80	2 (3.4%)	7 (18%)	9 (9.1%)	
Sex				
Male	38 (64.4%)	26 (66.7%)	64 (65.3%)	
Female	21 (35.6%)	13 (33.3%)	34 (34.7%)	
Cardiovascular disease	3 (5%)	5 (13%)	8 (8.1%)	
Chronic respiratory disease	3 (5%)	6 (15.4%)	9 (9.2%)	
Cerebrovascular disease	0	5 (13%)*	5 (5.2%)	
Diabetes	2 (3.4%)	9 (23%)*	11 (12%)	
Malignancy	1 (1.7%)	1 (2.5%)	2 (2%)	
Dementia	0	1 (2.5%)	1 (1%)	
Smoker	6 (10%)	8 (20%)	14 (14.2%)	
Notes.

* p < 0.05 compared to survival group.

Imaging of COVID-19 critically ill patients with pneumonia

The CT scan findings showed ground-glass opacity (GGO) in the subpleural distribution, combined with interlobular septal thickening and air bronchus sign (Figs. 1–4). Early manifestations included unilateral or bilateral limited inflammatory infiltration of the lungs, commonly displayed as subpleural patches, masses, segmental or subsegmental GGO. This may be accompanied with vascular congestion and thickening. There were multiple manifestations of ground glass shadow, consolidation shadow, intralobular septal thickening, and interstitial changes (Fig. 1).

Figure 1 A 68-year-old male COVID-19 patient.

(A–D) The chest CT scan showed an infection in both lower lungs. The main lesion is located under the pleura, with patchy and ground-glass opacity (GGO) manifestations. (E–H) Chest CT after 5 days. GGO and consolidation of bilateral lungs increased significantly. A large area of GGO in the left lung, with thickened interlobular septa, showing a “paving stone signs”.

Symptoms that occurred later were manifested by increased lesions and an enlarged scope. Bilaterally, there were multiple lobes of both lungs that were involved with some marked lesions. GGO coexists with real variable shadows, presenting “paving stone signs” or combined with fibrotic lesions and bronchial inflation (Fig. 2). The critical stage is characterized by diffusing lung lesions. GGO progresses with “paving stone signs”. This is an important feature of the COVID-19 pneumonia consistent with normal viral pneumonia, indicating the major involvement of the virus in the interlobular septum. The critical stages may result in the distribution of “white lung”. Thus, the lungs are dominated by consolidation shadows, combined with ground glass shadows, bronchial inflation, and could be accompanied by pleural effusion or pleural adhesions. However, swollen lymph nodes are rare (Fig. 3). During the remission period, lesions in both lungs can be gradually absorbed and improved with visible fiber strands (Fig. 4).

Figure 2 A 68-year-old female COVID-19 patient.

(A–D) CT scan of the chest showed infection of both lower lungs. The main lesion is located under the pleura, with patchy appearance and ground-glass opacity (GGO) manifestations. The chest CT scan showed “white lungs” with signs of bronchial inflation.

Figure 3 A 70-year-old male patient with COVID-19.

(A–D) Chest CT scan showed ground-glass shadows of the upper and lower lobes of both lungs. (E–F) Chest CT after 5 days. The ground glass shadow and consolidation of both lungs were visible. Chest CT showed “white lungs” with signs of bronchial inflation.

Figure 4 A 56-year-old male COVID-19 patient.

(A–D) Chest CT scans showed multiple ground-glass opacity (GGO), consolidation and fibrous cord lesions in both lungs. (E–H) The patient’s symptoms were significantly relieved 15 days after treatment. Chest CT showed that most of the lesions in both lungs were absorbed.

Symptoms, complications, and treatment

The main clinical symptoms of critical COVID-19 patients are illustrated in Table 2 and Fig. 5. Of the 98 critically ill patients, 85 (86.7% of the total) had a fever; 82 (83.7%) had a cough; and 53 (59.2%) had dyspnea. COVID-19 patients can be diagnosed by chest CT scan five days after the onset of symptoms. The median duration from symptom onset to admission to ICU is 10 days. Among the critically ill cases, 39 patients (39.8%) developed hospital-acquired infections, and 10 (10.2%) developed bacteremia (Table 2). The common commodities included 48 cases (49%) of ARDS, 36 cases (37.7%) of acute kidney injury, 32 cases (32.6%) of MODS, 21 cases (21.4%) of cardiac injury, 16 cases (16.3%) of liver dysfunction, and two cases (2%) of pneumothorax (Table 2 and Fig. 6). The median score of acute physiological and chronic health assessment II (APACHE-II) for all patients is 20. The median troponin was 0.50 ng/ml (Table 3). Hospital acquired pneumonia (HAP), gastrointestinal hemorrhage, hyperglycemia, vasoactive medications, and continuous renal replacement therapy (CRRT), and blood purification treatment were all markedly higher in the death group compared to the survival group (Table 2). Furthermore, patients with chronic diseases were prone to die from COVID-19 infection (Table 1 & Fig. 7).

Table 2 Clinical characteristics, comorbid conditions and treatment of patients with confirmed COVID-19.

	Survival (n = 59)	Death (n = 39)	Total (n = 98)	
Signs and symptoms				
Fever	47 (80%)	38 (97.4%)*	85 (86.7%)	
Cough	47 (80%)	35 (89.7%)	82 (83.7%)	
Dyspnea	14 (24.7%)	39 (100%)*	53 (59.1%)	
Myalgia or fatigue	12 (20%)	16 (41%)*	28 (28.6%)	
Arthralgia	2 (3.4%)	3 (7.7%)	5 (5.1%)	
Chest pain	0	2 (5.1%)	2 (2.0%)	
Headache	2 (3.4%)	4 (10%)	6 (6.1%)	
Vomiting	2 (3.4%)	2 (5.1%)	4 (4%)	
Diarrhea	3 (5.0%)	4 (10%)	7 (7.1%)	
Comorbid conditions				
ARDS	9 (15.2%)	39 (100%)*	48 (49%)	
MODS	2 (3.4%)	30 (76%)*	32 (32.6%)	
Acute kidney injury	6 (10%)	30 (76%)*	36 (37.7%)	
Acute cardiac injury	6 (10%)	15 (40%)*	21 (21.4%)	
Acute liver injury	6 (10%)	10 (25.6%)*	16 (16.3%)	
Hyperglycemia	8 (13.5%)	12 (30%)*	20 (20%)	
Gastrointestinal hemorrhage	1 (1.7%)	3 (7.7%)*	4 (4.0%)	
Pneumothorax	1 (1.7%)	1 (2.5%)	2 (2.0%)	
HAP	6 (10%)	33 (84.6%)*	39 (39.8%)	
Bacteremia	0	10 (25.6%)*	10 (10.2%)	
Urinary tract infection	0	2 (5.1%)	2 (2.0%)	
Treatment				
High-flow nasal cannula	10 (17%)	0*	10 (10.2%)	
Invasive ventilation	5 (8.4%)	39 (100%)*	44 (44.9%)	
Non-invasive ventilation	44 (74.5%)	39 (100%)	83 (84.7%)	
Prone position	1 (1.7%)	9 (24.6%)*	10 (10.2%)	
ECMO	0	1 (2.5%)	1 (1.0%)	
CRRT	1 (1.7%)	19 (48.7%)*	20 (20.4%)	
Vasoactive medications	1 (1.7%)	38 (97.4%)*	39 (40%)	
Antiviral therapy	59 (100%)	39 (100%)	98 (100%)	
Antibiotic therapy	59 (100%)	39 (100%)	98 (100%)	
Use of corticosteroid	59 (100%)	39 (100%)	98 (100%)	
Use of immunoglobulin	40 (67.7%)	39 (100%)	79 (80%)	
Chinese medicine medications	50 (100%)	20 (51.3%)	70 (71.4%)	
Use of Vitamin C	41 (70%)	35 (90%)	76 (77.5%)	
Enteral nutrition	59 (100%)	39 (100%)	98 (100%)	
Notes.

ARDS Acute Respiratory Distress Syndrome

MODS Multiple Organ Dysfunction Syndrome

HAP Hospital Acquired Pneumonia

ECMO Extracorporeal Membrane Oxygenation

CRRT Blood Purification Treatment

* p < 0.05 compared to survival group.

Intensive care measures and laboratory results between the survival and death groups of COVID-19

Forty-four patients (44.9%) had invasive mechanical ventilation; 83 patients (84.7%) received noninvasive mechanical ventilation; 10 patients (10.2%) needed a high nasal flow cannula; and 10 patients (10.2%) were treated with prone ventilation. One case (1%) underwent extracorporeal membrane oxygenation (ECOM). Twenty cases (20.4%) had renal replacement therapy; six cases (6.1%) were treated with blood perfusion; and six cases (6.1%) received plasma exchange. Five patients (5.3%) need convalescent plasma infusion, and 38 patients (38.7%) had vasoconstrictor medications. All the patients (100%) were treated with antimicrobial therapy, antiviral therapy, and glucocorticoid therapy. Moreover, 76 patients (77.5%) received high-dose vitamin C therapy; 70 patients (71.4%) received traditional Chinese medicine decoction.

Among 98 patients with critical illness of COVID-19, 39 (39.8%) died within 28 days of admission. The median time from admission to death in the death group was 12 days. Compared with the survival group, the death group was suffering more from ARDS (39 (100%) vs. nine (15.2%)), was more likely to have MODS (30 (76%) vs. 2 (3.4%)) (Fig. 7) and was given more invasive mechanical ventilation (39 (100%) vs. 5 (8.4%)).

As shown in Table 3, the oxygenation index (the ratio of oxygen partial pressure (PaO2) to FiO2) was lower in the death group (78 ± 18.2 vs. 115 ± 20.8 mmHg). Prothrombin time (s) was higher in the death group compared to the survival group (12.8 ± 2.9 vs. 26.8 ± 5.8). The duration from onset to ICU was longer and the heart rate was higher in the death group when compared with the survival group. In addition, the systolic blood pressure, the oxygenation index, the white blood cell count, and the prealbumin were all markedly lower in the death group when compared to the survival group. At the time of ICU admission, the death group’s APACHE-II scores, and the Sequential Organ Failure Assessment (SOFA) score were higher than the survival group’s scores. The lymphocyte count, the prothrombin time, the D-dimer, the lactate dehydrogenase (LDH), the creatine kinase isoenzyme-MB (CK-MB), the hypersensitive C-reactive protein (HS-CRP) were all higher in the death group when compared to the survival group. Furthermore, the survival rate of critically ill patients decreased with age (Table 1 and Fig. 8).

Figure 5 The main clinical symptoms of severe COVID-19 patients.

Figure 6 Comparison of comorbidity distribution between the death group and the survival group with severe COVID-19.

ARDS, acute respiratory distress syndrome; MODS, Multiple Organ Dysfunction Syndrome; HAP, Hospital Acquired Pneumonia.

Table 3 Intensive care measures and laboratory results of COVID-19 severe patients in both survival and death groups.

Intensive care measures and laboratory results	Survival (n = 59; ±SD )	Death (n = 39; ±SD)	
Duration from onset to imaging diagnosis (days)	5 ± 3.2	5 ± 2.6	
Duration from onset to ICU (days)	9 ± 2.3	12 ± 2.9*	
Heart rate (beats/min)	88 ± 22	100 ± 16*	
Systolic blood pressure (mmHg)	120 ± 26.7	100 ± 22.8*	
Oxygenation index (mmHg)	115 ± 20.8	78 ± 18.2*	
APACHE-II score on day 1	14 ± 2.2	19 ± 3*	
APACHE-II score	13.74 ± 3.46	22.91 ± 2.58*	
SOFA score on day 1	4 ± 2	8 ± 2*	
SOFA score	4.05 ± 2.48	8.48 ± 2.64*	
Hemoglobin (g/L)	121 ± 20	118 ± 14	
White blood cell count	5.44 ± 3.15	4.26 ± 2.66*	
Lymphocyte count (×109/L)	0.64 ± 0.34	0.58 ± 0.31*	
Platelet count	158 ± 76	198 ± 65	
Prothrombin time (s)	12.8 ± 2.9	26.8 ± 5.8*	
D-dimer (mg/L)	5.8 ± 2.1	9.5 ± 3.8*	
Albumin (g/L)	32 ± 12	30 ± 10	
Prealbumin (g/L)	160 ± 68	120 ± 51*	
Total bilirubin (μ mol/L)	12.5 ± 2.9	22.5 ± 12.1	
Serum creatinine (μ mol/L)	71.8 ± 25.4	120.4 ± 56.1	
LDH (U/L)	285 ± 102	392 ± 142*	
CK-MB (U/L)	56 ± 21	89 ± 38*	
Troponin (ng/ml)	0.6 ± 0.2	0.98 ± 0.3*	
Procalcitonin (ng/ml)	1.6 ± 0.5	5.8 ± 2.1*	
Interleukin-6 (pg/ml)	20.6 ± 9.1	25.8 ± 11.6*	
HS-CRP (mg/L)	5.6 ± 2.1	10.5 ± 5.2*	
Notes.

LDH Lactate Dehydrogenase

CK-MB Creatine Kinase Isoenzyme-MB

HS-CRP Hypersensitive C-Reactive Protein

* p < 0.05 compared to survival group.

Figure 7 Percentage distribution of chronic diseases in the death group and survival group with severe COVID-19.

Figure 8 The survival rate distribution of severe COVID-19 patients for all age groups.

Discussion

COVID-19 has affected millions of people around the world (Wiersinga et al., 2020). The novelty of our article focuses on the early stages of the COVID-19 outbreak in Ezhou, a key city 80 kilometers from Wuhan with a population of over one million, which has been largely overlooked in the literature. We know that Wuhan is the site of the first reported COVID-19 outbreak, providing unique insights into the virus’s initial stages. Patient data from the Ezhou area could be studied as a neighboring city via a critical but under-explored perspective. We believe that filling this gap in the literature represents a significant contribution and forms the core novelty of our work. Our study has two crucial aspects: (1) the timing of the outbreak, which focuses on the period immediately following the initial wave in Wuhan, and (2) the proximity to Wuhan, where the virus had not yet undergone significant mutations, unlike in other cities such as Beijing and Shanghai. His focus on a period when the virus was still in its original Wuhan form, combined with the geographical context, offers valuable insights that have been largely absent from existing research. Our paper includes detailed patient data that provide a critical perspective on the virus’s progression in a region close to Wuhan, making our better understanding of the early epidemic phase both unique and significant.

Specifically, we examined 98 critically ill patient information, which was taken from the early days of the COVID-19 spread. We were particularly interested in this crucial period since it might be possible to gain insights into how to react to virus attacks in the early days of the pandemic. The investigation of patterns in treatment options, lab values, most frequent symptoms, and imaging manifestations have been summarized in Fig. 9. We focused on the 39 individuals (39.8%) that died within 28 days of hospital admission. Forty-four (44.9%) of the critically ill patients required invasive mechanical ventilation and 48 (49%) had ARDS. During the beginning of the COVID-19 pandemic, there was no effective medicine and so the commonly used treatment was symptomatic support therapy (Yu et al., 2020). For patients in the mild and moderate stages, close follow-up may be sufficient to control this disease (Lai et al., 2020), while for patients in the heavy stage, active treatment and careful care may be adequate (Zheng et al., 2020).

Figure 9 Schematic displaying the clinical characteristics and imaging manifestations of 98 critically ill patients with COVID-19, in addition to illustrating the major findings of the death group in comparison to the survival group.

GGO, ground glass opacities; RRT, renal replacement therapy; HFNC, high flow nasal cannula; CCRT, continuous renal replacement therapy; AKI, acute kidney injury; ARDS, acute respiratory distress syndrome; HAP, hospital-acquired pneumonia; MODS, multiple organ dysfunction syndrome; ACI, acute cardiac injury.

Similar to SARS-CoV and Middle East Respiratory Syndrome (MERS-CoV), COVID-19 is a coronavirus that can be transmitted to humans, and these viruses are associated with high mortality in critically ill patients (Acter et al., 2020). Of the cohort of 38 severe SARS patients from 13 Canadian hospitals, 29 (76%) required mechanical ventilation, 13 (43%) patients died at 28 days, and six (16%) patients were still in mechanical ventilation. Out of a cohort of 52 severe COVID-19 patients from a hospital in Wuhan, China, 37 (71%) received mechanical ventilation therapy, of which 30 (81%) died in 28 days. In our cohort, the mortality rate is 39.8%, which is lower than that of the critically ill COVID-19 patients as mentioned above. The basic pathophysiology of severe viral pneumonia is severe ARDS. In the early days of the COVID-19 outbreak, the mortality rate from COVID-19 with ARDS was high; this was due to delayed diagnosis, shortage of medical resources, and limited treatment methods (Gibson, Qin & Puah, 2020; Grieco et al., 2020). The mortality rate decreased after standardization, homogenization, and individualization of medical treatment.

Near the beginning of COVID-19, there were more male patients than female patients (Sun et al., 2020). In addition, we noticed that the death group was older than the survival group. Based on previous research, older male patients are most susceptible to the COVID-19 virus, which is consistent with our figures (Sun et al., 2020). The study showed that COVID-19 patients with cerebrovascular disease are at increased risk of death, and we found that to be true in our analysis of the Ezhou Central Hospital information (Table 1). Additionally, prothrombin time (PT) measures how long it takes for a patient’s blood to clot and is an essential indicator of coagulation function. In our COVID-19 patients, an abnormal (prolonged) PT shown in the death group signals coagulation dysfunction and a hypercoagulable state can be likely associated with the disease’s severe forms (Table 3).

In our cohort, fever is the most common symptom of COVID-19. However, not all patients had a fever (15 cases (15.3%)), which made it difficult for identifying the presence of COVID-19. The median duration from symptom onset to pulmonary imaging diagnosis was 5 days. The disease progressed rapidly, and it manifested as diffuse lesions of both lungs, GGO combined with consolidation and “paving stones sign”, often accompanied by fiber cord performance. More than 80% of critically ill patients developed lymphopenia. The COVID-19 virus particles destroy the cytoplasmic components of lymphocytes, making lymphopenia a significant feature of COVID-19-infected patients. In addition, lymphopenia is commonly found in severely infected patients with MERS, as a result of lymphocyte apoptosis (Li et al., 2020a).

Mechanical ventilation is the main treatment for critically ill patients. The ratio of using mechanical ventilation between the survival group and the death group was significantly different, indicating that the oxygenation index was related to the severity and prognosis of the disease. In our study, barotrauma only occurred in two (2%) patients who had been hospitalized for nearly one month. Among SARS patients, about 25% of mechanically ventilated patients suffer from barotrauma (Gomersall et al., 2004). In our cohort, the lower incidence of barotrauma may be related to the widely accepted lung protective ventilation strategy employed in China. Also, the prone position and the extracorporeal membrane oxygenation (ECMO) were used to treat critically ill patients. All patients received two or more antiviral drugs and more than 90% of patients received intravenous glucocorticoid therapy. Although patients with severe SARS or with MERS pneumonia are usually given intravenous glucocorticoids, there is debate about whether this approach is effective in treating COVID-19 infection (Russell, Millar & Baillie, 2020).

Furthermore, this study from the Ezhou cohort, demonstrated a significant mortality rate in critically ill COVID-19 patients, particularly among the elderly and those with complications such as ARDS and MODS. The prevalence of ground-glass opacities and interlobular septal thickening on CT scans indicates potential fibrotic changes. As shown in a previous study, the 6-month follow-up findings focused on monitoring and early interventions to reduce pulmonary damage. While pulmonary dysfunction gradually improves in some cases, fibrosis can worsen over time (Stoian et al., 2023). Previous study showed that mechanical ventilation raises the risk of pulmonary fibrosis and contributes to long-term lung dysfunction (Cabrera-Benitez et al., 2014). Also, our data suggests that early identification of severe disease with timely interventions are essential to improve outcomes. Dynamic monitoring, imaging assessments, and antifibrotic therapies are critical to addressing post-COVID-19 complications effectively.

On the positive side, vaccination has played an important role in mitigating the COVID-19 pandemic. While the early stages of the outbreak, as seen in Ezhou, were affected by high mortality rates, severe complications, and heavy dependence on mechanical ventilation, the introduction of vaccines altered the course of the pandemic. Vaccination decreases the severity of infections and mortality, particularly in vulnerable populations such as the elderly with comorbidities. Our results suggest that through limiting viral transmission and reducing the burden on healthcare systems, vaccines facilitate a global recovery, helping patients from severe complications like ARDS and MODS. Continued vaccination efforts remain crucial in preventing future outbreaks and reducing the global impact of evolving COVID-19 strains.

Thrombotic and microthrombotic changes were often underemphasized contributors to mortality in severe COVID-19 cases. Critically ill patients frequently exhibited hypercoagulability, associated with complications such as pulmonary embolism, deep vein thrombosis, and diffuse microvascular thrombosis. Research showed that these events were triggered by cellular signals. Pulmonary microthrombi were found to be involved in ARDS, resulting in hypoxia and deterioration (Potpara et al., 2023). Thus, early identification of thrombotic risks are essential for reducing mortality.

We understand that antiviral therapies were limited at the start of the COVID-19 pandemic, and their effectiveness in severe cases was poor. Patients were often administered lopinavir/ritonavir and oseltamivir, initially used based on their success with other coronaviruses like SARS and MERS. However, these had minimal impact on viral replication or patient outcomes. Remdesivir later emerged with modest benefits, particularly in reducing hospitalization duration, but it was unavailable in the pandemic’s early phase (Indari et al., 2021).

In critically ill COVID-19 patients, antibiotic therapy was routinely administered due to the high risk of secondary bacterial infections and hospital-acquired infections, such as ventilator-associated pneumonia. Early uncertainty of COVID-19’s progression is often associated with empirical antibiotic use. However, overuse can cause the emergence of both multidrug-resistant (MDR) and extensively drug-resistant (XDR) bacteria, which complicate treatment and outcomes. Furthermore, prolonged antibiotic exposure increases the risk of opportunistic infections like Clostridioides difficile (Schechner et al., 2021).

The early spread of COVID-19 in Ezhou shows how healthcare resources are allocated in areas with limited medical infrastructure. Ezhou, which is a small city, often faces shortages of facilities and advanced treatment options. By analyzing the clinical characteristics of critically ill patients in Ezhou, this study helps understand the gap between urban and suburban pandemic responses.

It is recommended that early intervention strategies, including rapid case identification and efficient patient triage, are particularly relevant for future pandemic preparedness, as they provide a framework for improving response strategies in areas vulnerable to emerging infectious diseases. Understanding the trajectory of COVID-19 in a smaller city helps refine predictive models for disease spread, severity, and healthcare demands in non-urban areas.

Understanding the clinical characteristics of critically ill COVID-19 patients provides a foundation for retrospective analyses of predictive models for patient outcomes. Our results from Ezhou, where the virus remained in its original form before any potential mutations, provide a baseline for evaluating disease progression, severity markers, and treatment options. By comparing early-stage imaging patterns—such as ground-glass opacities and interlobular septal thickening—with later cases involving mutated variants, researchers can assess how the virus evolved with different strains related to disease severity. Therefore, as new variants of SARS-CoV-2 continue to emerge, we can analyze past patient outcomes about viral mutations and treatment interventions helps refine therapeutic strategies and optimize healthcare responses.

As a neighboring city just 80 kilometers from Wuhan, we can see that Ezhou represents an important yet underexplored setting that provides critical view into how the virus spread and impacted communities outside of the initial epicenter. Unlike Wuhan, which had extensive medical infrastructure and research focus, smaller cities faced distinct challenges in terms of healthcare capacity, resource allocation, and patient management. Thus, our study aims to fill the gap by providing detailed analysis of critically ill COVID-19 patients in Ezhou during the early pandemic, shedding light on disease progression in a city with more limited medical resources.

From our results there is a valuable contrast to studies conducted in larger cities. Understanding the trajectory of COVID-19 in a smaller city like Ezhou is important for refining predictive models of disease spread, severity, and healthcare demands in non-urban areas. The study in Ezhou also provides a retrospective view of COVID-19’s progression for future public health planning in similar urban environments. While early studies concentrated on Wuhan’s patients, this research provides a complementary perspective on how the virus affected critically ill patients in a nearby but less well-documented location, with a more nuanced understanding of COVID-19’s initial spread, a valuable reference for both historical epidemiology and future pandemic responses.

Fortunately, since 2024, we can tell that the mortality rate for COVID-19 has plummeted to 5% or less, and the current concern gradually shifts to the prevention of secondary superinfections, such as ventilator-associated pneumonia and nosocomial respiratory infections caused by bacteria or viruses (Klompas, 2024). Compared to patients on ventilators without COVID-19 infection, ventilated COVID-19 patients are twice as likely to be infected with pathogens. While the bacteria pathogens such as Pseudomonas aeruginosa, Klebsiella pneumoniae, and Staphylococcus aureus are similar between COVID-19 and non-COVID-19 patients, COVID-19 patients are more likely to have aspergillosis infection and herpes virus reactivations (Vacheron et al., 2022; Gioia et al., 2024). These secondary superinfections are associated with prolonged mechanical ventilation, which is the cornerstone treatment for the most concerning complications for COVID-19, acute respiratory distress syndrome (ARDS) that may exacerbate developing acute respiratory failure. From the recent findings, critically ill COVID-19 patients may further develop pulmonary fibrosis, which is associated with high plasma level of surfactant protein D, a marker for epithelial damage (Pan et al., 2025). While it was known that older age, disease severity, and increased level of inflammatory cytokines are correlated with development of pulmonary fibrosis in COVID-19 patients, one recent study discovered male gender and higher initial Sequential Organ Failure Assessment (SOFA) score were also correlated with pulmonary fibrosis, whereas SOFA score is used to assess the rate of organ failure for critically ill patients (Kim et al., 2024).

Currently, multiple randomized clinical trials (RCT) and meta-analyses were done to alleviate the symptoms and reduce the mortality rate for COVID-19 patients with ARDS. One study evaluated the use of “triple therapy” (corticosteroids, therapeutic plasma exchange, and timely intubation with lung ventilation) has shown to markedly increase survival for patients with COVID-related sepsis and ARDS (Keith et al., 2024). Surprisingly, gender, age, and comorbidities are not the strongest predictor for COVID-19 patient mortality; increased level of interleukin 6 (IL-6), C-reactive proteins (CRP), and fewer lymphocytes have been shown to be significantly associated with mortality, while decreased levels of IL-6 and white blood cell count 1–2 days after ICU admission are significantly associated with survival (Bartoszewicz et al., 2024). Another recent study has also evaluated the inflammatory parameters of 30-day mortality, and both high ferritin and high APACHE scores were identified as strong predictors of post-admission mortality (p < 0.001) in receiver operating characteristic (ROC) curve (Demirel & Miniksar, 2024). For COVID-19 patients experiencing a subsequent cytokine storm, an H-score, represented by reactive hemophagocytic lymphohistiocytosis (HLH) diagnosis score, could suggest potential secondary HLH, which is a positive score associated with increased mortality rates (Oppenauer et al., 2025). Taken the most recent emerging studies together, the current data have shown a more thorough characterization about the improved outcome of critically ill COVID-19 patients with updated treatments. As ongoing collaborations among international clinical trial groups have offered valuable insights to determine the degree of interventions based on COVID-19 symptom severity, more research is still needed for effective evidence-based medicine by reconsidering differential diagnoses and treatments for each patient individually (Ling et al., 2024).

Conclusions

Our data show that critically ill COVID-19 patients exhibited a high mortality rate accompanied by significant and diverse imaging changes. Early and repeated imaging features hold crucial clinical value for the timely diagnosis and management of COVID-19. Most patients suffering from severe comorbidities relied heavily on mechanical ventilation. Elderly subjects, such as those over 60 years old or with complications, potentially had a significant risk of mortality. The functional CT scans could provide useful findings in the lower bilateral lung lobes and help understand disease progression with potential intervention strategies. This study focuses on Ezhou, a city near Wuhan that was critical during the early pandemic. It fills a gap in understanding the virus when it remains in its original format. The findings provide healthcare workers with essential tools to recognize early symptoms with imaging patterns, helping with prompt treatment to reduce mortality and alleviate severe outcomes.

Supplemental Information

Supplemental Information 1 Raw Data

We thank the Chinese National Health Commission for coordinating data collection for patients with COVID-19 infection. We also thank H Ye, S Zuo, W Jiang, J He, Y Zhang, F Strickland, J Dickson, and K Sipprell for their assistance during the manuscript preparation.

Additional Information and Declarations

Competing Interests

Author Contributions

Human Ethics

Data Availability

Li Zuo is an Academic Editor for PeerJ.

Guohui Yang conceived and designed the experiments, performed the experiments, analyzed the data, prepared figures and/or tables, authored or reviewed drafts of the article, and approved the final draft.

Zewen Liu conceived and designed the experiments, performed the experiments, analyzed the data, prepared figures and/or tables, authored or reviewed drafts of the article, and approved the final draft.

Tabitha Abraham analyzed the data, prepared figures and/or tables, authored or reviewed drafts of the article, and approved the final draft.

Linwei Li analyzed the data, authored or reviewed drafts of the article, and approved the final draft.

Tingyang Zhou analyzed the data, prepared figures and/or tables, authored or reviewed drafts of the article, and approved the final draft.

Qing Zhang performed the experiments, analyzed the data, authored or reviewed drafts of the article, and approved the final draft.

Li Zuo conceived and designed the experiments, analyzed the data, prepared figures and/or tables, authored or reviewed drafts of the article, and approved the final draft.

The following information was supplied relating to ethical approvals (i.e., approving body and any reference numbers):

Ethics Committee of the Ezhou Central Hospital (IRB#L2020-Y-012).

The following information was supplied regarding data availability:

The raw data are available in the Supplemental File.

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
