# Peer review of "Unveiling the clinical profiles of critically ill COVID-19 patients: insights from Ezhou during the early spread"

_PeerJ, doi:10.7717/peerj.20318_

## Round 0.1 · original submission · Major Revisions

· Academic Editor

Major Revisions

Authors should highlight the novelty of the study and discuss in the finding in depth.

Reviewer 1 ·

Basic reporting

The paper provides no new knowledge that would have a practical use, especially in 2024 under the dominance of a completely different virus lineage (also in terms of clinical severity), nor does it provide a novel view on past experiences with COVID-19. Oddly, the authors refer only to studies published in 2020 (with two citations to papers from 2021). The manuscript is not well fit in the current understanding of COVID-19 and its clinical evolution.

Experimental design

no comment

Validity of the findings

The article reports on the clinical characteristics of critically ill COVID-19 patients from the early stages of the SARS-CoV-2 epidemic. My main concern is related to the value of the study in 2024 in terms of better understanding COVID-19. The clinical characteristics of COVID-19 patients, including those with more severe disease and critically ill, have been extensively reported over the last few years. The findings presented in the submitted manuscript would provide important insights 3-4 years ago, not at the end of 2024. The results provide no novel perspective or understanding of disease. I disagree with the authors that their findings, at this stage of investigating COVID-19 all over the world, can provide useful information to medical doctors on how to recognize COVID-19 pneumonia. By 2024 this should be well known from clinical practice and profound research, and can be found not only in international but most of national recommendations regarding COVID-19. The authors also state in the Introduction that no study addressed clinical or imagining characteristics of critically ill COVID-19 patients in the early stages of the pandemic, which is false.

There are numerous studies that did, and below, please see 3 examples that took me not even 5 minutes to find:
- https://doi.org/10.1007/s00330-020-06955-x
- https://doi.org/10.1097/rli.0000000000000689
- https://doi.org/10.1148/radiol.2020200370
-

Even in 2020, clinical and chest imaging features of COVID-19 were already the subject to reviews and meta-analyses, e.g.,
- https://doi.org/10.1016/j.mayocp.2020.10.022
- https://doi.org/10.21037%2Fatm-20-2124

I am very sorry, but I cannot recommend this manuscript for publication as it provides no new knowledge that would have a practical use, especially in 2024 under the dominance of a completely different virus lineage (also in terms of clinical severity), nor does it provide a novel view on the past experiences with COVID-19.

Reviewer 2 ·

Basic reporting

Clear, unambiguous, professional English language used throughout.
Introduction and background to show context are relevant.
References in the literature are poor considering the explosion of articles written on this subject
The structure conforms to PeerJ standards, then can be observed especially in the part of tables and figures.
Figures are relevant, high quality, well labeled and described, comparable to literature data
The data are mostly explained.

Experimental design

In the discussion part, these could be well explained
The research is original but does not bring new elements. It is a review of the principles, symptoms, clinical manifestations, evolution and main complications of the Covid-19 Disease

The research does not fill a knowledge gap, but comparing it to other viral respiratory diseases, eg MERS, it is easy and pleasant.
Fibrotic lung lesions must be discussed more exhaustively, they were only mentioned, see the fibrotic changes in Covid patients, (Long-Term Radiological Pulmonary Changes in Mechanically Ventilated Patients with Respiratory Failure due to SARS-CoV-2 Infection. Biomedicines 2023

Validity of the findings

The dominant thrombotic and microthrombotic changes in patients with severe forms of Covid-19 were also not mentioned, being an important cause of death ( COVID-19 Thrombotic Complications and Therapeutic Strategies)
The antiviral therapy administered at the beginning of the Covid 19 pandemic was quite poor. It would be good to clarify which antivirals were administered to these patients, especially in the severe forms that required mechanical ventilation.
To clarify if these patients were anticoagulated or not?? considering the increased risk of thrombotic complications.
In the discussions, it must be explained the reason why all patients received antibiotic therapy and the risk of its routine administration without evidence, such as the development of MDR, xdr germs or Clostridoides difficile infection (COVID-19 and Clostridioides difficile Coinfection Analysis in the Intensive Care Unit)
Let's not forget the role that vaccination had in extinguishing this pandemic, which continues to make me victims on a global level

Annotated reviews are not available for download in order to protect the identity of reviewers who chose to remain anonymous.

Reviewer 3 ·

Basic reporting

1.1 The introduction fails to highlight the novelty of the study. Several studies on COVID-19 imaging and clinical profiles have been published, and the authors should better articulate how their findings fill a distinct knowledge gap.
1.2 Some figures are of low quality (e.g., figure 1, 3, and 5) and lack important details like scale bars (e.g., figure 5-8). Ensure all abbreviations are defined in figure legends and tables.

Experimental design

2.1 This manuscript lacks how this study adds to the existing literature in the introduction.
2.2 The study mentions retrospective analysis but does not discuss potential selection biases in patient inclusion (e.g., referral bias to the designated hospital for critical patients).

Validity of the findings

3.1 Conclusions are supported by the data but lack broader contextualization. Discuss how findings could guide treatment strategies or resource allocation during pandemics.

Additional comments

Line 32: Replace "Wuhan We analyzed" with "Wuhan. We analyzed."
Line 32: "36 miles)"—Ensure consistent use of metric (e.g., kilometers instead of miles, unless journal style requires miles).

---

## Round 0.2 · Minor Revisions

· Academic Editor

Minor Revisions

Please revise the manuscript according to the reviewer's comments.

Reviewer 3 ·

Basic reporting

The manuscript was revised in this part.

Experimental design

The manuscript was revised in this part.

Validity of the findings

This part needs to be revised.

Additional comments

The revised manuscript successfully emphasizes the study's unique geographical and temporal focus—Ezhou, located near Wuhan, during the early stages of the COVID-19 pandemic. However, the novelty argument can be strengthened to make the contribution more compelling. Currently, it leans heavily on the uniqueness of the study period and location, which, while valid, may not fully resonate with readers in 2024.
Improvement Suggestions:
1. Broaden the Contextual Relevance: Highlight how studying the early pandemic in a smaller city like Ezhou provides insights into disease progression and resource allocation in regions with limited healthcare infrastructure. This can bridge the gap between urban and suburban pandemic responses. Discuss how findings from this study could inform models for future outbreak preparedness, particularly in areas prone to emerging infectious diseases.
2. Draw Connections to Modern Relevance: Articulate how understanding the clinical and imaging characteristics of critically ill patients during the early stages can guide retrospective analyses or improve predictive models for patient outcomes. Emphasize how this historical data could serve as a baseline for comparing the evolution of the disease and the impact of subsequent mutations or interventions.
3. Address Current Research Gaps: While early COVID-19 studies have been abundant, the manuscript can highlight the scarcity of in-depth analyses specific to small urban centers like Ezhou during the pandemic's onset. This helps underscore its contribution to a more nuanced understanding of the virus's initial spread.

---

## Round 0.3 · Minor Revisions

· Academic Editor

Minor Revisions

Majority of the comments have been addressed. However, please update the introduction and the references. Then compare your results with the findings from more recent publications.

Reviewer 2 ·

Basic reporting

Dear authors,
I have reviewed the manuscript titled "Unveiling the Clinical Profiles of Critically Ill COVID-19 Patients: Insights from Ezhou During the Early Spread." Below are my comments and recommendations following the journal's guidelines:
The manuscript is clear and easy to read, and the text is understandable without any issues. However, I noticed that all the references cited are from 2020, which is quite dated. Consider-ing the extensive body of literature published after that period, I believe it is essential to update the references to include more recent studies. There has been a significant amount of research published since the early stages of the pandemic, and it is important to reflect the current state of knowledge in this area.

Experimental design

The design of the study is solid and appropriate. However, to ensure the manuscript aligns with the latest scientific knowledge, it would be beneficial to update the information provided in light of recent findings in the literature. This update is essential to maintain the manuscript's relevance and accuracy

Validity of the findings

The results accurately reflect the reality of the early stages of the pandemic, and the data pre-sented were highly valuable at that time. However, it is crucial to compare and discuss these results in the context of more recent publications (2024-2025). This will help demonstrate how the clinical profiles of critically ill COVID-19 patients have evolved over time, considering the advancements in understanding the disease and improvements in patient care.

Additional comments

The study is well-conceived, clearly written, and analyzes the early period of the COVID-19 pandemic, focusing on the characteristics of SARS-CoV-2 infection. The data presented were certainly of high interest in 2020, but given the wealth of information available since then, an update is necessary. To make this article publishable, I strongly recommend that the authors revise the manuscript to incorporate the most recent findings from 2024-2025. This includes revising the discussion section to reflect current knowledge on COVID-19, particularly regard-ing complications such as COVID-19-related pneumonia, mechanical ventilation, pulmonary fibrosis, and infectious complications in critically ill patients.
In conclusion, the manuscript presents valuable insights, but it requires significant updates to be considered for publication. I encourage the authors to revise the manuscript by incorporating recent literature and addressing the current state of knowledge on the disease.
Sincerely,

Reviewer 3 ·

Basic reporting

no comment

Experimental design

no comment

Validity of the findings

no comment

Additional comments

Having reviewed the revised manuscript and response, I suggest acceptance. The authors adequately addressed the suggestions, enhancing the manuscript's novelty and value. It now meets publication standards.

---

## Round 0.4 · accepted · Accept

· Academic Editor

Accept

The introduction has been updated with the inclusion of more recent references.